# Registered Report: Neural correlates of thematic role assignment for passives in Standard Indonesian

**Bernard A. J. Jap**[1]*, **Yu-Yin Hsu**[2], **Stephen Politzer-Ahles**[3]

**1** Department of Humanities, Language and Translation, School of Arts and Social Sciences, The Hong Kong Metropolitan University, Hong Kong, **2** Department of Chinese and Bilingual Studies, The Hong Kong Polytechnic University, Hong Kong, **3** Department of Linguistics, University of Kansas, United States of America

* bajjap@hkmu.edu.hk

## Abstract

Previous studies conducted across multiple languages have found processing differences between patient-first and agent-first word orders. However, the results of these studies have been inconsistent as they do not identify a specific event-related potential (ERP) component as a unique correlate of thematic role processing. Furthermore, these studies generally confound word order with frequency, as patient-first structures tend to be infrequent in the languages that have been investigated. There is evidence that frequency of syntactic structure plays an important role in language processing. To address this potential confounding variable, we test Standard Indonesian, a language where passive structures occur with high frequency and are comparable in frequency to active structures. In Standard Indonesian, there is evidence from acquisition, corpus, and clinical data indicating that the use of passive is frequent. In the present study, 60 native speakers of Indonesian read 100 sentences (50 active and 50 passive) while EEG was recorded. Our findings reveal neural correlates of thematic role processing in the passive sentence condition – specifically, a positive shift corresponding to a P600 on the verb, and a more sustained positivity on the second noun phrase. These findings support existing evidence that sentences with a 'non-default' word order impose increased cognitive load, as reflected by ERPs, even when they occur with higher frequency in the language.

## Introduction

For most individuals, understanding sentences is an effortless, uneventful process. However, early research has suggested that not all sentences are equal during language comprehension. One example is the passive structure. Compared to active sentences, passives are acquired and used later by children [1], take longer time to process [2], and are prone to misinterpretation by adults [2]. Additionally, passive

**Data availability statement:** All relevant data and analysis script are publicly available on the Open Science Framework (OSF) repository through the following link: https://osf.io/vhdz2/.

**Funding:** This research is funded by The Hong Kong Polytechnic University (UGC), PolyU Dean's Reserve (ZVXM) and CBS research fund (ZZRX). YYH, SPA, and BAJ are recipients of the grant. URL: https://www.polyu.edu.hk/rio/ The funders did not have a role in study design, data collection and analysis, decision to publish, or preparation of the manuscript.

**Competing interests:** The authors have declared that no competing interests exist.

sentences also pose challenges for individuals with brain lesions [3]. Debates regarding how humans process different types of sentences have evolved over time. The use of event-related potentials (ERPs) to study language processing has become common not only among language researchers, but also among psychologists and clinicians.

In recent years, the vast number of studies investigating neural responses during language processing have mainly focused on linguistic anomalies. Different types of linguistic violations have yielded distinct ERP responses. The P600, for example, is seen in response to syntactic violations such as violating the expected word order [4] or morphosyntactic inflection [5], reflecting semantic integration in the context of semantic illusions [6], and the processing of long-distance wh-dependencies [7]. Other ERPs, such as the N400, appear as a result of anomalies such as semantic incongruence [8], the integration of the overall sentence meaning [9], or memory retrieval [6].

## Word order processing in ERP research

While the study of ERP responses to linguistic anomalies has been valuable, it's also important to investigate how the brain processes different grammatical structures that are typical but vary in their characteristics. Specifically, research comparing neural responses to different grammatical word orders [10] has shown less consistency and often faces challenges from potentially confounding variables such as syntactic frequency. This subsection discusses studies that have focused on word order effects in grammatical sentence processing.

To better interpret our findings in the context of current neurolinguistic theories, it's important to understand the functional roles attributed to the N400 and P600 components in the context of sentence comprehension. The N400 component is traditionally associated with semantic processing, but contemporary accounts increasingly interpret it as reflecting lexical-semantic retrieval processes rather than semantic integration [6]. According to the Retrieval-Integration [11] account, N400 amplitude reflects the ease with which conceptual knowledge associated with a stimulus can be retrieved from long-term memory.

The P600 component, initially linked to syntactic violations, is now understood more broadly as reflecting mental representation construction and revision [12]. In the context of passive structures, disambiguation toward a passive reading would elicit a P600 effect for several reasons: First, passive structures involve non-canonical thematic role assignments requiring representational updating - the first noun phrase must be integrated as a patient rather than the default agent role [7]. Second, Aurnhammer et al. [13] argued that the P600 reflects integration difficulty for structures that deviate from expected patterns, regardless of whether they contain anomalies. Third, the P600 reflects the increased processing demands associated with updating mental representations when encountering unexpected linguistic input [14].

These word order effects can be understood through the lens of incremental processing theories, where readers assign thematic roles (agent, patient) to noun phrases as sentences unfold. Across diverse languages, processing systems appear

to employ an "agent-first" preference as a default strategy [15]. This S/A preference represents a neurophysiological bias to initially interpret noun phrases as agents, even in languages where this might require later revision. When this initial preference is contradicted, Haupt et al. [16] argued that the reanalysis process involves increased costs in "linking" arguments from syntactic to semantic representations, rather than phrase structure revisions. This distinction manifests in different ERP components: while phrase structure revisions typically elicit P600 effects, grammatical function reanalyses often correlate with N400 effects or biphasic N400-late positivity patterns.

Some studies have documented differences in the processing of different word orders, such as between subject and object relative clauses [10,17], as well as of simple sentences such as the subject-object and the object-subject word order distinction in German [18]. Most of these studies have investigated languages with a canonical agent-first word order, either agent-patient-verb or agent-verb-patient, and have reached similar conclusions: Patient-first structures require more effort to process than agent-first structures. For example, in the sentence with an object-embedded relative clause (1a), "The man" is not the agent of the first verb, thereby this does not adhere to the typical English word order in which the agent is placed in the first position [19]. This results in a predicted increase in syntactic processing demand (as reflected in reaction times or accuracy rates, see 2), compared to a sentence with a subject-embedded relative clause (1b) which follows the agent-first expectation.

| 1a) The man who the woman violently scolded admitted the error. |
| --- |
| 1b) The man who violently scolded the woman admitted the error. |

Matzke et al. [18] compared object-before-subject (OS) to subject-before-object (SO) structures in German with the provision of case information through the use of articles. In addition, they also examined sentences in which case was temporarily ambiguous, by using feminine case markers on the article (*die*) of the NP1 which, in German, can signify both the accusative and the nominative case (2 and 3).

| 2) Object - Subject | | | | | |
| --- | --- | --- | --- | --- | --- |
| Die | begabte Sängerin | entdeckte | der | talentierte | Gitarrist. |
| The $_{(Fem\ Nom/Acc)}$ | gifted singer | discovered | the $_{(Masc.Nom)}$ | talented | guitar player |
| *'The talented guitar player discovered the gifted singer.' Ambiguous until 'der'* | | | | | |

| 3) Subject - Object | | | |
| --- | --- | --- | --- |
| Die | begabte Sängerin | entdeckte | den | talentierte Gitarrist. |
| the $_{(Fem\ Nom/Acc)}$ | gifted singer | discovered | the $_{(Masc.Acc)}$ | talented guitar player |
| *'The gifted singer discovered the talented guitar player.' Ambiguous until 'den'* | | | |

In the masculine NP1 sentences, for which case is unambiguous, Matzke et al. [18] found that the accusative NP1 (Patient-Agent order) elicited a Left Anterior Negativity (LAN) compared to the nominative NP1 (Agent-Patient order) and this effect continued for the rest of the sentence. Left fronto-temporal negativity was observed following the 2nd article in the Object-Subject condition. In the condition that was ambiguous up to the second article (as in examples 2 and 3), a P600 was found in the disambiguation section (the 2nd article 'der') of the Patient-Agent compared to the Agent-Patient structures. Matzke et al. [18] attributed the initial LAN on the NP1 to working memory. In a similar experiment, Schlesewsky, Bornkessel, and Frisch [20] found that the LAN was only observed in object-first non-pronominal NP1s in German, and not in pronominal NP1s (as shown in example 4).

| 4) Object-first pronominal structure | | | | | | | |
| --- | --- | --- | --- | --- | --- | --- | --- |
| Gestern | hat | ihn | der | Vater | dem | Sohn | gegeben. |
| yesterday | has | it $_{ACC}$ | the $_{NOM}$ | father | the $_{DAT}$ | son | given |
| 'Yesterday, the father had given it to the son'. | | | | | | | |

As such, they came to the conclusion that the LAN originates from a local syntactic mismatch via the violation of canonicity principles in non-pronominal NP1s, rather than from higher working memory usage due to dislocated objects in general.

Other studies which have investigated thematic role assignment processing through ERPs included Meltzer and Braun [10], who compared the processing difference between subject-embedded and object-embedded relative clauses in English, and they found a negativity at NP1 (400–800ms) and a positive shift at the offset of the relative clause for reversible clauses. Jackson, Lorimor, & van Hell [21] observed positivity at the 500–700ms time window when comparing English passive structures to active structures; this positivity was strongest at the left anterior electrodes. In a study of Japanese, Wolff, Schlesewsky, Hirotani, and Bornkessel-Schlesewsky [22] included sentences that were similar to those in German, in which the patient-first structures were compared to agent-first structures via the use of a suffix in the nominative or accusative case on the NP. ERPs for patient-initial compared to agent-initial structures after NP1 included an early (120–240ms) negativity, which was referred to as 'scrambling negativity'; a broadly distributed positive shift at the NP1 of Patient-Agent structures (400-650ms); an N400 at the NP2 for Agent-Patient structures; and a late parietal negativity (650–1050ms) at the verb. Aside from the object scrambling, the positive shift at the NP1 in patient-initial sentences was interpreted as the resolution of dependency introduced by an accusative-first argument. Another study on Japanese [23] investigated how context usage influences processing of patient-first structures. They observed that patient-first structures elicited a sustained LAN at NP1 and P600 at NP2 when the NP1 was new and not provided in the context.

A similar late parietal negativity at the verb was reported in Japanese in another study that used scrambled sentences [24]. Erdocia et al., using simple declarative sentences in Basque [25], which marks NPs with the ergative and absolutive cases, showed a similar negativity post-onset of the NP1 of patient-first sentences. Basque, like Japanese, is verb-final and allows for both agent-first and patient-first orders of the NPs. At the NP2 position, a left negativity (400–550ms) for patient-first structures was found. In the P600 time window (700–900ms) in the verb position, a parietal positivity was observed. The negativity at the NP1 for patient-first structures, although observed in a different time window, is suggested to be related to the scrambling negativity found in both German and Japanese. The effect at the NP2 is interpreted as a LAN that expresses working memory usage for displaced elements, or, alternatively, the authors suggested that agents and patients are processed differently regardless of their position. They hypothesized that the P600 observed at the verb position for patient-first structures relates to an increase of processing costs when elements are displaced from their canonical positions. These studies attributed the late negativity to general increased processing of scrambled sentences.

For case-marking languages, there appears to be a pattern of a negative shift in NP1, which is the critical region of the sentence for thematic role assignment, for patient-first as compared to agent-first orders. However, other parts of the sentence indicate a different pattern: The P600 found in both Basque [25] and English [21] verbs is somewhat contradictory, as the case information denoting the thematic role assignment was provided earlier at the NP1 in Basque; as such, this P600 cannot be attributed to thematic role processing, and the authors attributed it to 'higher syntactic complexity'. By contrast, a study of Japanese [22] found a late negativity on the verb, which was an effect in a similar time window but with opposite polarity. Due to the inconsistencies in these findings (which may be in part attributed to the experimental design and the languages examined in these studies), it is particularly difficult to draw a conclusion on what constitutes a neural correlate of thematic role processing. To summarize the findings of comparable studies, Table 1 outlines the results of the discussed literature with results from [10] presented separately because, unlike the other studies, they discuss relative clauses with different regions measured.

### Some relevant properties of Standard Indonesian

Indonesian is a zero-marking language [26] without case or gender markings. Transitive verbs are usually only inflected for voice (active or passive), and there is no verbal inflection for tense, aspect, or agreement. Indonesian has SVO word order [27]; however, the ordering of constituents can be flexible, and it is possible (although infrequent) for verbs to take

**Table 1. Summary of previous ERP studies comparing word orders.**

| Language | Conditions | NP1 | NP2 | V | Note |
|---|---|---|---|---|---|
| German* [18] | Agent-verb-patient vs patient-verb-agent Ambiguity (fem. NP1 vs masc. NP1) | LAN (400–600ms, 600–800ms) | - Negativity (400–1000ms) <br> - P600 (600–800ms, 800–1000ms) for amb. fem. NP1 | Not discussed | Nom/Acc case was provided by articles preceding NPs. |
| Japanese* [22] | Agent-patient-verb vs patient- agent-verb | - Scrambling negativity (120–240ms) <br> - positivity (400–650ms) | N400 (300–500ms) | Late negativity (650–1000ms) | Nom/Acc case was provided by markers following the NPs. |
| Basque* [25] | Agent-patient-verb vs patient- agent-verb | Negativity (300–500ms) | Negativity (400–550ms) | P600 (700–900ms) | Erg/Abs case was provided by markers following the NPs. |
| English* [21] | Agent-verb-patient vs patient-verb-agent | Not discussed | Not discussed | P600 (500–700ms) | Frontal distribution of P600- different from the typical distribution in garden-path sentences. |
| | Conditions | NP1 | RC onset | RC offset | |
| English [10] | S.RC – O.RC Reversibility | Negativity (400–800ms) for reversible (i.e., ani. NP1 vs inani. NP1) | Not found | positivity (-300–100ms) for rev. conditions | Only found reversibility effects, no word order effect. |

*all components are evoked to compare patient-first to agent-first structures

the initial position. Chung [28] suggested that Indonesian belongs to a branch of the Austronesian language family that was originally verb-initial, as the passivized-transitive, active-transitive, as well as intransitive verbs can take the 1st position.

The usual transitive passive (5b) has the patient in the initial position. Examples of typical simple active and simple passive sentences are as follows:

5a) Simple active (agent-verb-patient)

| Perempuan | itu | **men**dorong | laki-laki | itu |
|---|---|---|---|---|
| girl | the | **ACT**-push | boy | the |

'the girl is pushing the boy'

5b) Simple passive (patient-verb-agent)

| Laki-laki | itu | **di**dorong | (oleh) | perempuan | itu |
|---|---|---|---|---|---|
| boy | the | **PAS**-push | (by) | girl | the |

'the boy is pushed by the girl'

The sentence (5a) contains a verb with an active-transitive voice marking (*men*-). Many base verbs in Standard Indonesian (SI), like in (5a) go through nasal substitution [29] in the active voice, a common process (see S1 for the stimuli list), which substitutes the root–initial obstruent for one of the nasal sounds n, ŋ, or m. Likewise, the passive is expressed by the prefix (*di*-) on the verb in (5b), where 'the boy' is the patient of the action. Similar to English, the *by*-phrase is optional in the passive. In addition, the preposition *oleh* 'by' may be omitted when the agent is immediately adjacent to the verb [30]. As the current study observes the ERP distinctions between the simple active and the simple passive, it is worth noting that the typical passive in Indonesian, unlike in most Indo-European languages, is highly frequent. It is acquired at a very early stage (around 2 years old [31]) compared to English (4–5 years old), which can be attributed to its high input frequency of 28–35% in Indonesian, compared to 4–5% in English. While actives remain more common overall, the

active-passive frequency difference in Indonesian is much smaller than in previously studied languages. This difference is also reflected in the written form: Only 9% of English verbs display passive morphology [32] compared to 30–40% of Indonesian verbs [33] having the passive marker 'di-'. A typologically notable feature of Standard Indonesian is that agentless passives (where the agent is entirely omitted) are extremely common, comprising up to 86.6% of passive tokens in some corpora, while the agentive passives used in this study make up only 2.7% to 16.5% of passive tokens [34]. This frequency pattern suggests Indonesian speakers initially expect an agentless passive when encountering a passive verb. Consequently, additional integration processes may be required when an agent is encountered.

## The present study

The current study investigates the processing of non-anomalous, simple sentences with different word orders. The focus of this study is on the critical point in time during sentence comprehension, which is mainly the verb, but another region of the sentence (such as NP2) are also investigated to check for spill-over effects.

The two conditions are the active and the passive sentences. The materials are typical, plausible, reversible (both noun phrases are animate), violation-free sentences. There are certain aspects of the Indonesian language that make the topic of the present study worth pursuing. First, the thematic roles of the NPs are coded by the passivization prefix on the verb, rather than by case marking on the NPs. Second, unlike the studies of German [18] (for feminine nouns in NP1s) and Basque [25], no ambiguity manipulations are involved in the critical region. Third, the structures to be tested in the current study are both relatively frequent compared to the passive structures in previously studied languages and are considered to be typical. The patient-first conditions in the previous studies are infrequent in the respective languages compared to their agent-first counterparts; for example, the object-embedded relative clause in English [35] and the patient-agent structure in German [36] are both highly infrequent structures. While arguments against exclusively syntactic frequency-based accounts of sentence processing have been established for German [37], there is behavioral evidence of the influence of sentence-level frequency and its interaction with other syntactic contrasts, such as the lexical bias of verbs [37–39].

There have been recent studies that directly address the relationship between frequency and thematic role processing. Huber et al. [40] used language model-derived surprisal values across German, Hindi, and Basque to predict ERP patterns, finding that surprisal alone cannot capture N400 effects without incorporating an explicit agent preference principle. Similarly, Sauppe et al. [41] investigated Äiwoo, a language with patient-first default order, revealing that human referents were still preferentially interpreted as agents despite contradicting statistical patterns in the language. Isasi-Isasmendi et al. [42] found N400 effects in Basque when ambiguous arguments were disambiguated toward patient readings in intransitive verbs. These studies collectively suggest that both frequency-based expectations and general agent preference mechanisms contribute to thematic role processing. Our study extends this line of inquiry by examining a language where the 'non-canonical' passive structure is relatively frequent, allowing us to more effectively isolate the effect of word order from frequency effects.

The current experiment contributes electrophysiological data from Standard Indonesian, a zero-marking language that lacks many morphosyntactic features of Indo-European languages, paired with a relatively rigid word order. To the best of our knowledge, no ERP study on word order effects has investigated Indonesian, where the 'non-canonical' sentence structure is relatively frequent. This is important because all the previous studies have compared one common sentence structure to an extremely infrequent one, and there is a wealth of evidence suggesting that syntactic frequency plays a role in sentence processing at the behavioral level [2,43,44].

## Method

### Participants

Our data were recorded from 60 adult native speakers of Indonesian. Of these, data from 9 participants were excluded due to low trial counts after artifact rejection. This left data from 51 participants (2 male, mean age= 35.2) who had at least

25 trials in every condition. All the participants were tested using an Indonesian translation of the short form of the Edinburgh Handedness Inventory [45] to ensure that they are classified as right-handed. Participants provided informed consent and were financially compensated. As a power analysis is not possible due to lack of detailed data in previous studies including variance across trials within a participant, we used a heuristic estimate of the sample size needed by doubling the sample size of a relevant study on non-anomalous sentence processing: Jackson, Lorimor, & van Hell [21] had 25 participants, and the present study recruited 51 participants. The participants completed a questionnaire to ensure that they have first-language or equivalent proficiency in Indonesian. This study was approved by the Hong Kong Polytechnic University's Institutional Review Board (ref no. HSEARS20211223003). The recruitment period for this study started at 06 November 2022 and ended at 01 March 2023.

The protocol for this study was approved as a Registered Report Protocol [46]. The present study includes several deviations from the Registered Report Protocol (Table 2. Our original criteria for participant exclusion were the following: having 25 trials per conditions (50% of the maximum trial number), a signal-to-noise ratio significantly above 3 dB (based on the method by Parks et al. [49]), no more than 15% bad channels, no more than 6 bad channels in one cluster/adjacent to one another, and a set of completed questionnaire and demographic information. During data collection, however, we found that this set of criteria was very conservative – only 16 of our participants met this inclusion threshold – and was more strict than comparable studies. Therefore, we chose to relax the criterion; we report an additional analysis of the smaller sample meeting the original conservative inclusion criteria in the Supporting Information (S3 Appendix in S2 File). Below, we have included a summary of the list of deviations from the registered report protocol (Table 2).

## Materials

Participants read stimuli comprising 100 semantically reversible active and passive sentences (see S1 Appendix in S1 File), examples of which are shown in Table 3.

The sentences involve an animate agent and patient and were constructed while avoiding a plausibility bias. Word order is manipulated by using two structures: simple active and simple passive sentences. Based on previous studies on processing well-formed sentences, the contrast of interest is passive-active, and whether the effects of the word order persist. We strive for active-passive pairing plausibility with the use of NP1 and NP2 that are plausibly reversible (i.e., each sentence should not be strongly biased to one interpretation due to the nouns used). Additionally, there are 200 filler trials consisting of questions (e.g., *What did the adventurer notice yesterday?*) and cleft sentences (e.g., *It is the boy who the girl is calling*) to prevent the habituation of the participants. The task for both the experimental and filler sentences is discussed in the Procedure section. To ensure that the nouns were equally plausible as agent and theme, the stimuli were tested for acceptability and prototypicality in an online survey method (n=21) using a Likert scale of 1–7 (1 being highly

**Table 2. List of deviations from the registered report protocol.**

| Section | Description and justification |
|---|---|
| Exclusion criteria | The initial proposed set of exclusion criteria appears to have been too conservative, excluding the vast majority of participants' data including many that we believe are relatively clean. As such, it has been relaxed to no longer include the signal-to-noise ratio criterion. However, we have included a separate analysis in S3 Appendix in S2 File with the subset of participants that were included according to the criteria of the registered report protocol. |
| Preprocessing | In the registered report protocol, we mentioned that the data will be bandpass filtered (0.1–40.0Hz) as a final preprocessing step. However, this did not seem to be the best practice for a preprocessing pipeline especially one that runs ICA. Instead, ICA decomposition is greatly improved with a prior high-pass filter [47,48]. As such, instead of doing all filtering at the end of preprocessing, we instead ran a 0.1 Hz high-pass filter *before* ICA, and a 40 Hz low-pass filter *after* ICA decomposition and removal of bad ICs. |
| Analysis | We added one-tailed tests on positive clusters, as positive clusters were observed on the two-tailed tests, which were initially the only planned test on the protocol. Both analyses yielded the same result. |

**Table 3. Stimuli examples of each condition.**

| Condition | NP1 | Art | VP | Adjunct | NP2 | PP/RC |
|---|---|---|---|---|---|---|
| Active | Wanita | itu | mewawancara | langsung | seorang pria | di kantor. |
| | Woman | that/the | **ACT**interview | immediately | (a)man | at office. |
| | (the/a) woman immediately interviews (the/a) man at the office. | | | | | |
| Passive | Wanita | itu | diwawancara | langsung | oleh pria | di kantor. |
| | Woman | that/the | **PAS**interview | immediately | by man | at office |
| | (the/a) woman is immediately interviewed by (the/a) man at the office | | | | | |

acceptable/extremely plausible and 7 being highly unacceptable/extremely implausible) with a similar setup used by Jackson, Lorimor, & van Hell [21]. Outside of acceptability of the stimuli, participants of this survey also rated the plausibility of two noun-verb-noun combinations (E.g. First noun as agent, *wanita mewawancari pria the woman interviewed the man*; second noun as agent, *pria mewawancari wanita the man interviewed the woman*). All materials were rated plausible and acceptable by native speakers of Indonesian with plausibility of the stimuli at 1.93 out of 7 (*SD* = 0.57) and acceptability of the stimuli at 2.32 (*SD*=0.44). The items were rated equally plausible regardless of whichever noun was presented first as the agent (first noun as agent: *M*=1.9, *SD*=.7; second noun as agent: *M*=1.98, *SD*=.76; Paired samples t-test between the mean scores: $t(49)=1.03, p=.31$).

The materials were presented visually and word-by-word for 500ms with a 100ms blank screen between each word. This presentation paradigm is typical of sentence processing experiments using rapid-serial-visual presentation (RSVP), and our word presentation time is actually longer than many comparable ERP studies to accommodate the potentially longer words in Standard Indonesian. For comparison, Jackson et al. [21] used 350ms per word with 100ms ITI for English; Matzke et al. [18] used 200ms per word with 300ms ITI for German; Wolff et al. [22] used 650ms with 100ms ITI for Japanese; and Erdocia et al. [25] used 250ms per word with 250ms ITI for Basque. While effects in later time windows (e.g., 500–700ms) occur when the word is no longer physically on screen, this is standard in ERP sentence processing research as neural responses continue after stimulus offset. Each trial began with a blank screen of 750ms followed by the word '*siap?*' (ready?), which stayed on the screen until the participant pressed any key. There were two lists of sentences each pseudorandomized into 10 sets (each set containing 10 experimental sentences and 20 fillers). Digital triggers were inserted at two time points in every sentence: at the onset of the verb (i.e., the onset of the prefix), and at the onset of NP2. Additionally, ERP epochs relative to NP1 were later extracted from -1400 to -200 before the verb onset to be used to check whether there are baseline differences between the conditions. The critical triggers are located in the middle of the sentence.

## Frequency

The main aim of this study is to essentially 'control' for the possible effect of frequency. It might not be feasible to manipulate the relative syntactic frequency of a structure in a language. However, at the lexical level, verb bias (or the frequency with which the verb appears in different structures) influences both production [50] and comprehension [51]. While most of the previous studies did not address this variable in their materials, one study by Jackson et al. [21] attempted this by using the progressive participle '-ing' in English (which is less frequent than the simple active verb form and is somewhat comparable to the passive) in combination with the past tense instead of simple present active structures. This has two potential issues; the first is that we interpret thematic role processing as the assignment of agent/patient role to the first/second noun phrase in the study, and the active voice has several verb forms such as the simple form in which the first noun phrase is the agent, and when considered as a collective as they should be, these occur more frequently than the passive form. The second issue is that it remains unknown whether the additional information from the progressive aspect, which deviates from the canonical simple present active, elicits a neural signal distinct from the simple present

active itself, especially when considered in the context of active versus passive comparisons. In our study, we attempt to control verb bias by incorporating verb pairs that fulfil either of the two criteria:

1. A verb with higher token frequency in its passive form compared to its active form.

2. A verb with the same frequency class* for both the passive and the active form.

   *Frequency class is a number assigned to a group of words whereby this number does not often change in different corpora. The calculation is as follows: The frequency of the most frequent word in the corpus is divided by the frequency of the specific word, and log base 2 of the result is rounded up to the closest whole number [52].

   We used the *Indonesian mixed corpus*, which is the largest Indonesian online corpus in the Leipzig Corpora Collection [52]. The distribution and descriptive information of the frequency can be seen in Fig 1 and Table 4, respectively (for full frequency information about the verbs, see S2 Appendix in S1 File). In Fig 2, log-transformed frequency values of passive verbs are subtracted from log-transformed frequency values of the active counterparts of the same verbs. The general pattern is not only are they relatively comparable, but if anything, the passive verbs are slightly more frequent for some pairs. Extremely infrequent words (with a frequency class of 20 or higher) were not used in this study, as the highest frequency class in our stimuli is 17 and the lowest is 7.

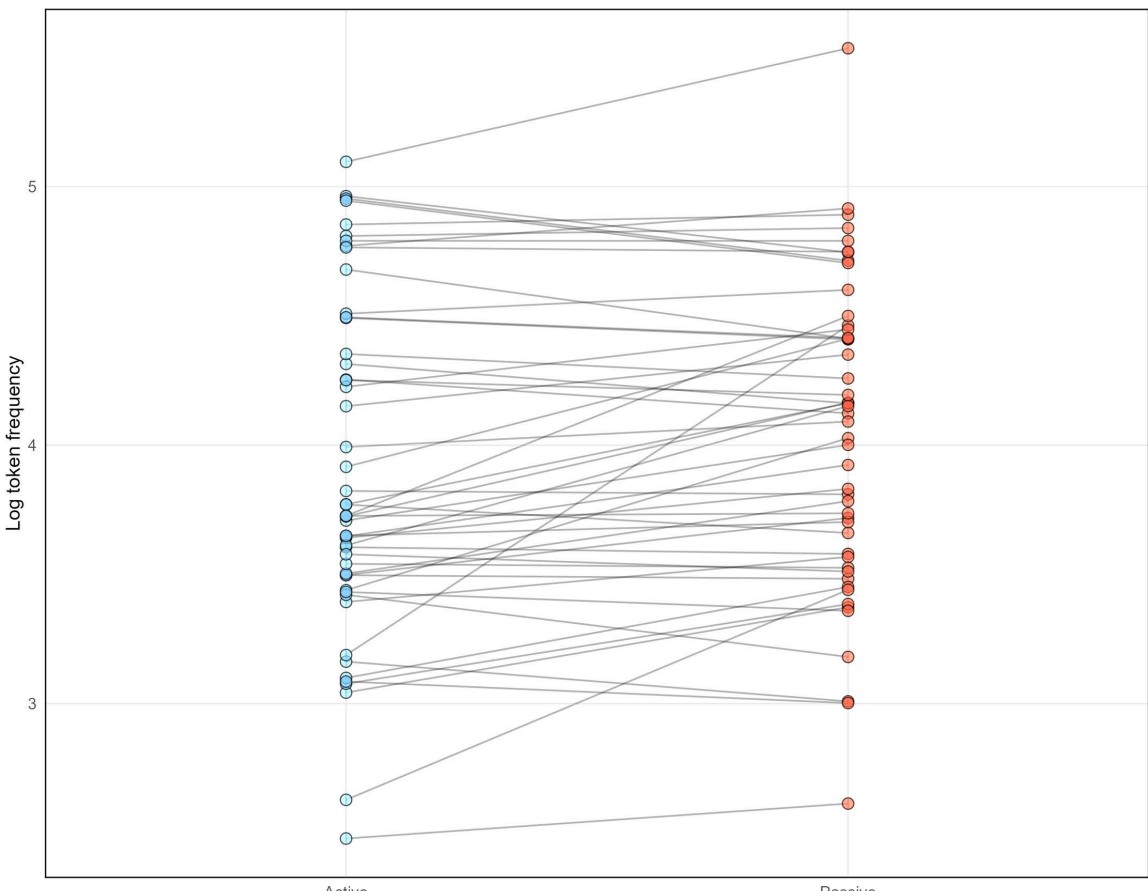

**Fig 1. Scatterplot of log-transformed passive and active verb frequencies.**

**Table 4. Descriptives of raw frequency for verb.**

|  | Active | Passive |
|---|---|---|
| mean | 21513.1 | 26415.88 |
| SD | 30110.56 | 50517.23 |
| *Range* | 124326 | 342184 |
| *Min* | 301 | 411 |
| *Max* | 124627 | 342595 |

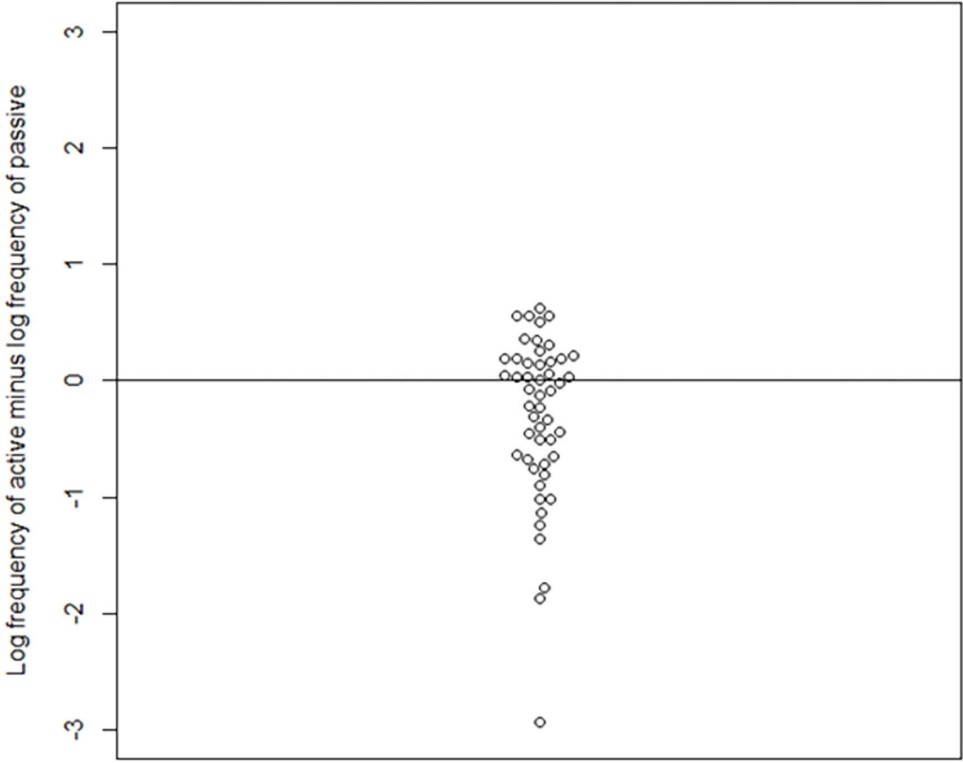

**Fig 2. Scatterplot of difference between active and passive (active – passive).**

## Procedure

Before the experiment, participants read an information sheet, filled out a questionnaire about their demographic details, and signed the informed consent form.

In each experiment, participants were seated in front of a presentation monitor on which the sentences would be presented using E-Prime software, and the experiment started with written instructions that were explained orally by the experimenter. Prior to the start of the experiment, participants were instructed to minimize their head movements during the trials and not to keep their eyes closed during the experiment, although they were not asked to refrain from blinking.

During the whole experiment, a fixation cross was shown between trials and sets. After every 3–8 trials (randomized), participants were given a comprehension prompt. For the experimental items and filler cleft sentences, the comprehension prompt is a yes/no question to probe the thematic role assignment (e.g., Did the police shoot the robber?/Did the robber shoot the police?). For the filler questions, the comprehension prompt is a sentence that appropriately responds to the

question (e.g., the adventurer noticed an officer patrolling) or if the noun phrase in the question is presented as the theme (e.g., the officer called the adventurer), and participants have to judge whether the sentence is an appropriate response to the question (e.g., they should respond "yes" if they see "the adventurer noticed an officer patrolling" after this filler). These prompts were designed to ensure that the participants remain focused and parse the trials displayed, as well as to provide a measure of comprehension performance.

To avoid fatigue, there were pauses after every block of 30 trials (10 blocks in total), where the participants could take a break and resume the experiment whenever they are ready. The total test session, including cap and electrode preparation, took approximately one and a half hours per participant.

## EEG recording and preprocessing

EEGs were recorded using 64 Ag-Acl electrodes that were attached to the participant's scalp via an elastic cap with a 10–20 system. Conductive gel was used. The cap had two dedicated electrodes for the left and right mastoids. To monitor horizontal and vertical eye movements, two electrodes were fixed in the outer canthi of each eye, and one more was placed below the left eye (the VEOG above the left eye is integrated in the cap). Electrode impedances were kept below 5 kΩ. The EEG was amplified and digitized with a sampling rate of 1000 Hz with an analog bandpass filter of 0.03–100Hz. The amplifier used was a SynAmps 2 (NeuroScan, Charlotte, NC, United States), and the cap was a 64-channel Quik-Cap Neo Net (NeuroScan, Charlotte, NC, United States). A Stimtracker (Cedrus) provided an interface between the experiment presentation software and EEG acquisition. Continuous EEG data were acquired using Curry 7 acquisition software (Compumedics NeuroScan) whereby the files were exported to the.cnt format and analysed using EEGLAB [53] for preprocessing, and FieldTrip [54] for the statistical analysis.

The EEG data were re-referenced to the average of the two mastoid electrodes. After interpolation of bad channels, the data were filtered using a 0.1Hz high-pass filter. The ERPs were calculated per participant, per electrode, and per condition in intervals of 200ms before onset to 1000ms after onset for each time-locked trigger. These epochs were then demeaned per channel in each epoch (the mean of the data from the entire epoch was subtracted from each data point, as this may result in better ICA decompositions than baseline-correcting based on pre-stimulus interval; [55]). The epochs were then be subjected to an independent component analysis using the runica() command in EEGLAB [56]; this divided the data into many independent components corresponding to the number of channels, excluding mastoid electrodes, EOGs, and bad channels that were previously marked. These components were inspected visually to identify blinks and saccades as well as muscle artifacts, and components that were identified to be associated with one of these types of artifacts were removed (a maximum of four components per participant were removed). After the removal of components, baseline correction was applied to the data with a 200 ms pre-stimulus onset baseline. Next, the epochs were run through a moving window peak-to-peak threshold function (window size of 200ms with a window step of 50ms and a threshold of 100μV) for artefact detection; epochs with artifacts were marked for removal based on this criterion. Then, the data were filtered using a 40.0Hz low-pass filter.

There is an alternative method of *not* using baseline corrections, as suggested by Wolff et al. [22] and Friederici et al. [57] who conducted sentence processing experiments. The reasoning behind this approach is that in the mid-sentence time windows, the waves of each trial may not be identical prior to the onset of the critical word, therefore potentially distorting the baseline. Wolff et al. [22] instead used narrower bandpass filters with higher low cutoffs (0.3–20.0Hz) to exclude slow drifts while still including language-related ERPs. However, Steinhauer [58] criticized the use of a higher filter instead of baseline-correction because first of all, the modified filter does not distinguish between artifacts (slow drifts) and real slow waves related to language processing; moreover, the filter converts sustained effects into apparent local effects such as ELANs, and finally, the increased filtering does not directly address the problem of a distorted baseline resulting from differences before the onset of the critical region. We therefore adopted a 200ms baseline in this study. It should also be noted that Alday [59,60] has proposed an alternative statistical approach to the baseline correction issue, suggesting

that the baseline period could be included as a predictor in a GLM analysis, allowing the data itself to determine the optimal amount of correction.

## Data exclusion criteria

Participants' data were excluded based on our predetermined data exclusion criteria, which include the following:

- Minimum number of trials after artifact rejection: Any participant having fewer than 25 trials in any condition after artifact rejection was excluded from the analysis. Here, 9 participants were excluded.

- Missing information/data: If a participant refuses to complete all or part of the questionnaire about her/his demographic information or the handedness inventory, or if a technical issue leads to missing data/a subset of missing data for a participant, the individual will be excluded from the analysis. No participant was excluded from this criterion.

- Bad channels (1): The threshold for data exclusion is at or over 15% of the electrodes (9 or more) being unusable due to excessive artifacts or environmental noise. This is in the event that these channels cannot be interpolated due to positioning (for example, multiple bad channels being adjacent to each other and therefore not having enough neighbouring electrodes for interpolation). No participant was excluded from this criterion.

- Bad channels (2): A second criterion is if a number of bad channels clustered or adjacent to one another: If there are 6 bad channels in one cluster/adjacent to one another, the participant was be excluded. No participant was excluded from this criterion.

## Statistical analysis

The planned statistical analysis was conducted using cluster-based permutation tests [60] over all the scalp electrodes and the entire post-stimulus epoch. The advantage of this approach is that it allows testing for effects anywhere on the scalp and any time in the epoch, while still controlling the familywise false positive rate, and without the experimenter needing to choose regions and time windows for analysis. The test works by comparing the ERPs elicited by the active and passive conditions at each channel and each sample, and identifying clusters of spatiotemporally adjacent data points where the difference between the two conditions exceeds some threshold; in our analysis, that threshold is two-tailed $p<.1$ in a t-test for the whole epoch. In other words, a t-test comparing active and passive were performed at every sample in every channel, and if a series of several time points in a row on the same channel and/or several adjacent channels at the same time all exceed this threshold, they are treated as a "cluster". Next, each cluster is assigned a test statistic (in our case, the test statistic for a cluster is derived by summing the $t$-values of all $t$he samples in the cluster), and the largest cluster-level test statistic in the epoch is taken as the observed test statistic for the data. Next, the data are randomly permuted (i.e., within each subject, the condition labels "active" and "passive" may be randomly switched) several thousand times, and with each random permutation the abovementioned procedure of identifying clusters and calculating a test statistic is repeated. This yields a permutation distribution of several thousand test statistics, against which the original observed test statistic is compared. The proportion of permutation test statistics that are larger than the original observed test statistic is the $p$-value for the test; if there is a significant difference between the active and passive ERPs then this value will be small.

Our rationale for using a cluster threshold of $p<.1$ was that this makes the test more sensitive to weak, sustained effects, similar to what was found in previous studies that have investigated thematic role processing. Effects were treated as significant if they obtained a $p$-value below .05 for the permutation test. The abovementioned cluster $p<.1$ criterion was only used in the procedure for generating clusters, not for actually evaluating the overall significance of the test. The $p$ threshold used for generating clusters does not affect the conservativity of the overall test, as explained by [60]. For any sample to be included in a cluster, it needed to have at least two spatial neighbouring electrodes that also meet the

threshold (we used the *minnbchan=2* function in the Fieldtrip implementation of the cluster-based test). The permutation tests used 5000 iterations.

## Predictions

There are several possible patterns of the results. These are described below.

1. An ERP contrast can be expected between active and passive sentences on the critical region (verb). Following the S/A preference hypothesis [15], we might expect a posterior positivity (P600) at the disambiguation point (the verb) reflecting revision of thematic information: NP1 being reassigned from the default agent role to patient. Alternatively, we might observe an N400 effect, reflecting increased "linking" costs rather than phrase structure revision [16].

2. An ERP contrast attributed to assignment of agent thematic role in a non-default position can be expected between active and passive sentences when participants read NP2. Additionally, based on earlier studies we expect a 'generalized' increase in processing costs in comprehending the passive structures attributed to violation of transitivity expectation [22] as well as retrieval of verbal material in a non-canonical position and uptick in working memory demands [25]. For example, the verbs in Basque [25] and Japanese [22] provide no additional morphosyntactic marking of thematic roles beyond what is already indicated by case markers, but within that section, ERP differences for patient-first compared to agent-first structures were observed.

3. Failure to observe an ERP correlate in both the verb and NP2, especially between the passive and active conditions, would suggest that previous findings regarding the neural correlates of thematic role assignment were confounded by syntactic frequency because the studies compared one highly frequent structure (e.g., active) to a highly infrequent sentence structure (e.g., passive).

In general, for the time windows and specific components, due to the nature of cluster-based permutation where adjacent sites and time points are correlated, we are looking at the whole epoch (here we theorize that a 'real' effect should persist through multiple adjacent electrodes and may be observable through a chunk of tens to hundreds of milliseconds/samples). However, were there to be an effect in either the verb or NP2 (as per the predictions above), we expect it to be observed in the time windows and distributions corresponding to the N400 and/or P600 – all of which have been observed in similar studies and above 300ms. Specifically, for the verb, we would expect the passive to be more positive than the active during the 500–700ms time window, as that is what the study most similar to this ours [21] reported in their critical region. For NP2, we expect some form of negativity to occur for the passive compared to the active between the 300–600ms time window – this was observed in studies which reported ERPs post-disambiguation.

## Results

Accuracy for the comprehension questions is at 70.67% (SD = 14.12). Figs 3 and 4 show representative channel ERPs at the verb and at NP2, respectively. Figs 5 and 6 show topographic plots of the effect during the 400–800ms time window at the verb and NP2. ERP plots of all 64 channels for all conditions and whole-epoch topographic plots are provided in the Supporting Information (S4 Appendix in S2 File).

The contrast between passive and active sentences generated a positivity (passive elicited a more positive ERP than active) around the time window of the P600 at the verb, while it elicited an earlier but sustained positive shift at the NP2. Both the positive ERPs at the verb and NP2 are left-lateralized and show a somewhat frontal distribution. This 'frontal' positivity around the P600 time window has also been reported in the reading of English passive compared to active sentences [21] and was noted to possess a distribution that is distinct from typical P600s elicited from, for example, garden-path sentences or syntactic violations.

Statistical analysis confirmed some of these observations. For the whole-epoch test, there is a significant positive difference between passives and actives in the NP2 region (*p*<.01), but this difference was not significant on the verb (*p*=.087).

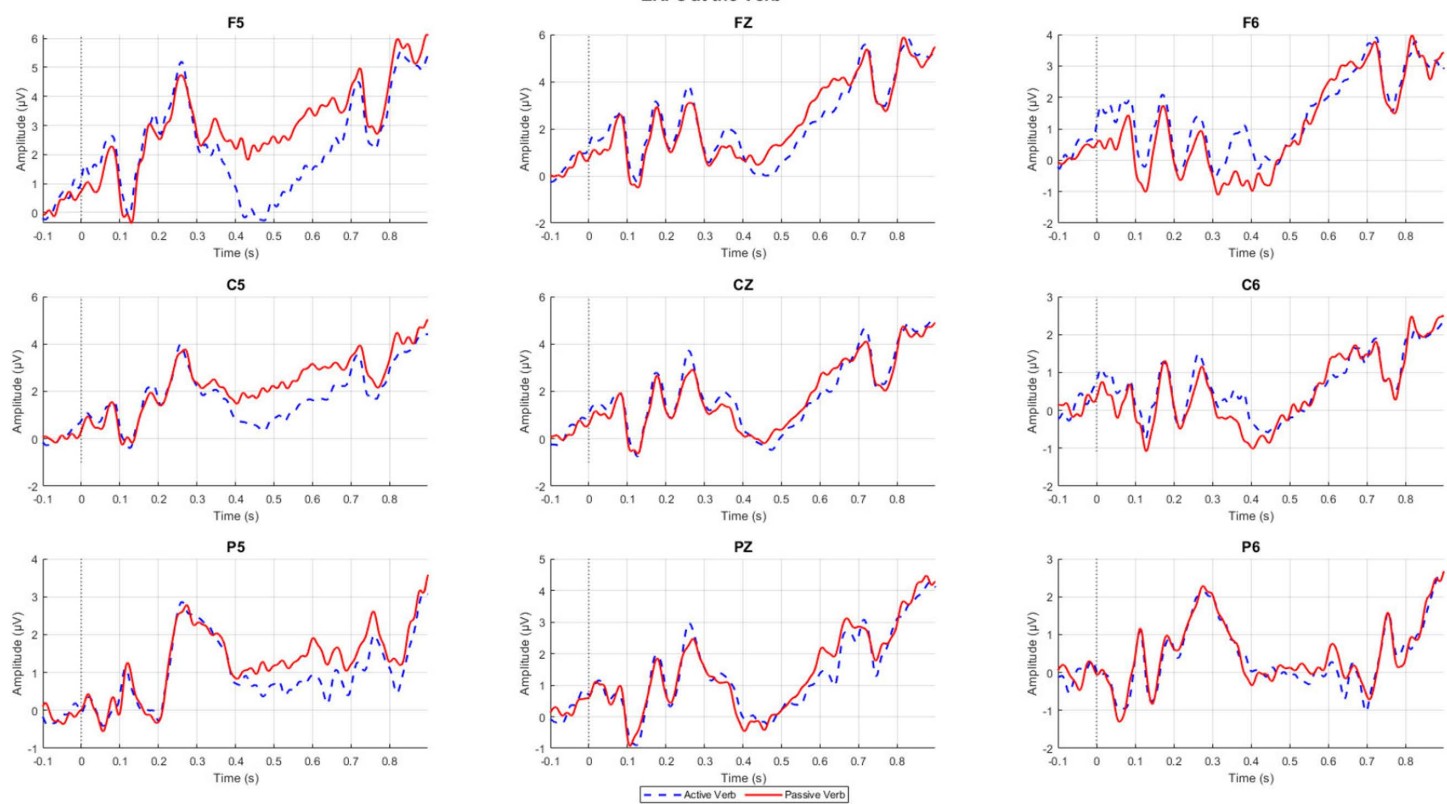

**Fig 3. ERPs at the verb in the following channels: F5, FZ, F6, C5, CZ, C6, P5, PZ, and P6.**

In an exploratory analysis focused just on the 500–700ms time window for the cluster test with the same specifications, this effect on the verb was observed to be significant ($p$=.039). While our pre-registered analysis found a significant effect at NP2, visual inspection of the spatiotemporal distribution of the statistical cluster that drove this effect suggests that this effect was also driven by differences that are in the early portion of the epoch. In other words, the passive-active contrast may have elicited an early and sustained positive shift that drove this statistical effect.

## Discussion

We presented an ERP experiment to test whether passive sentences require additional cognitive load to process compared to active sentences, despite having similar frequencies. For this, we turn to Standard Indonesian, an Austronesian language where the passive is comparatively frequent, salient, and acquired early, which resulted in relatively preserved and comprehension for brain-damaged individuals [61]. In such a case, we are able to isolate word order (or some form of syntactic operation such as movement) and thematic role assignment to test whether these processes do indeed generate increased processing costs through examining ERP correlates at the disambiguation region (verb) and after (NP2).

To summarize the findings in relation to our predictions, the verb and NP2 elicited a more positive ERP for passive than active sentences, mainly over left anterior channels. At the verb, this effect was weak and only emerged in a focused cluster-test on 500–700 ms post-stimulus; whereas at NP2 this effect was longer-lasting and more statistically significant. We did not find N400 or a similarly distributed effect on either the verb nor the NP2.

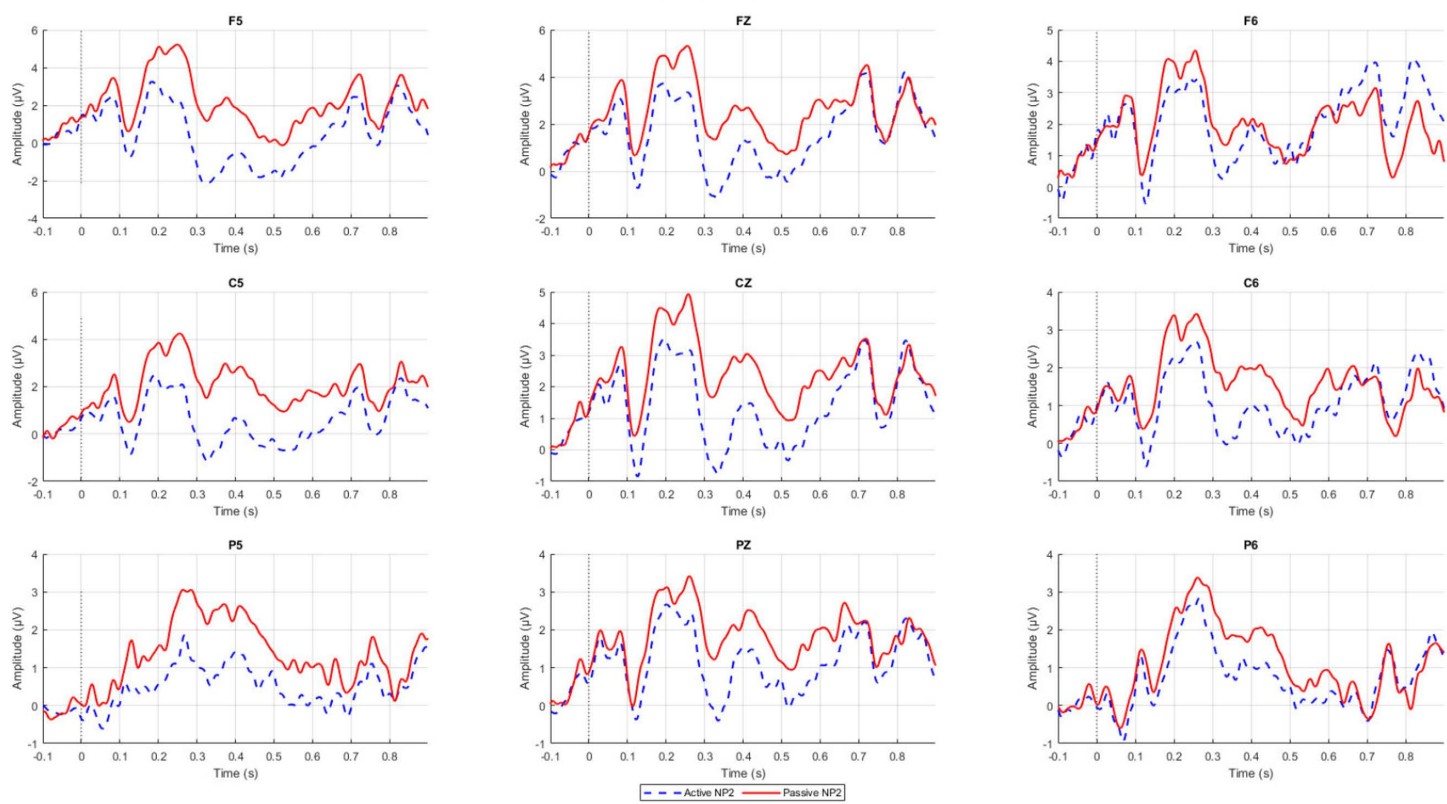

**Fig 4. ERPs at the NP2 in the following channels: F5, FZ, F6, C5, CZ, C6, P5, PZ, and P6.**

The ERP found on NP2 is somewhat similar to those found in Basque [25] and Japanese [22] post-disambiguation. While the NP2 itself may not provide the critical disambiguating thematic role information, readers still parse the NP2 as atypical in the sense that it is part of a non-canonical structure with the agent in the second position as well as transitivity violation in languages like Japanese [22]. Additionally, the NP2 ERP may be attributed to the fact that in Standard Indonesian, like in some other Austronesian languages (e.g., Javanese [34]), the agentless passive is the most frequent structure by an overwhelming margin. Therefore, it is reasonable to propose Standard Indonesian readers assume that they are reading an agentless passive at the point of the verb; however, when they were shown the NP2, participants would have to revise their previous predicted frame to integrate a structure that includes an agent. In a study on passive frequency [34], agentless passives range from 62.3% to 86.6% depending on the corpora type (narrative data, spontaneous speech, or written data), whereas the agentive passive, the type of passive we used for the stimuli, make up 2.7% to 16.5% of the tokens with the remaining passive type marked as 'abbreviated passive', a type of agentive passive where the oblique is not marked by a preposition.

While the spatial distribution of the positivity found on the verb may not correspond to the typical P600, it should be noted that the experiment tests well-formed sentences without any manipulation of grammaticality (i.e., no violations). Jackson, Lorimor, & van Hell [21] conducted two experiments, both comparing well-formed passive to active sentences, and they found a significant positivity on the P600 time window which, like our results, is maximally distributed in the anterior areas in both experiments. As such, our results are not entirely atypical keeping in mind the type of stimuli: the revision of thematic role assignment by the participants is similar to the processing efforts associated with higher syntactic

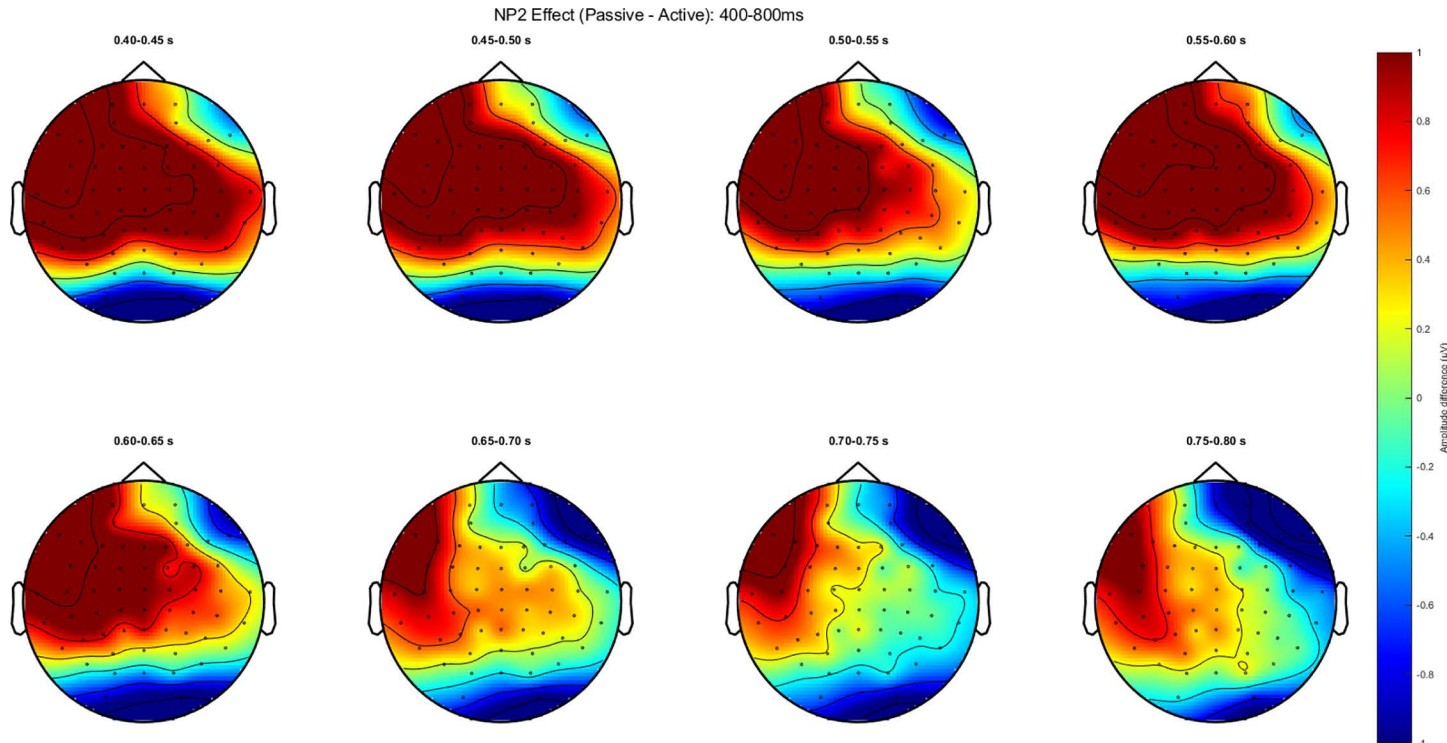

**Fig 5. Topographic plot at the Verb for the 400 to 800ms time window (passive-active).**

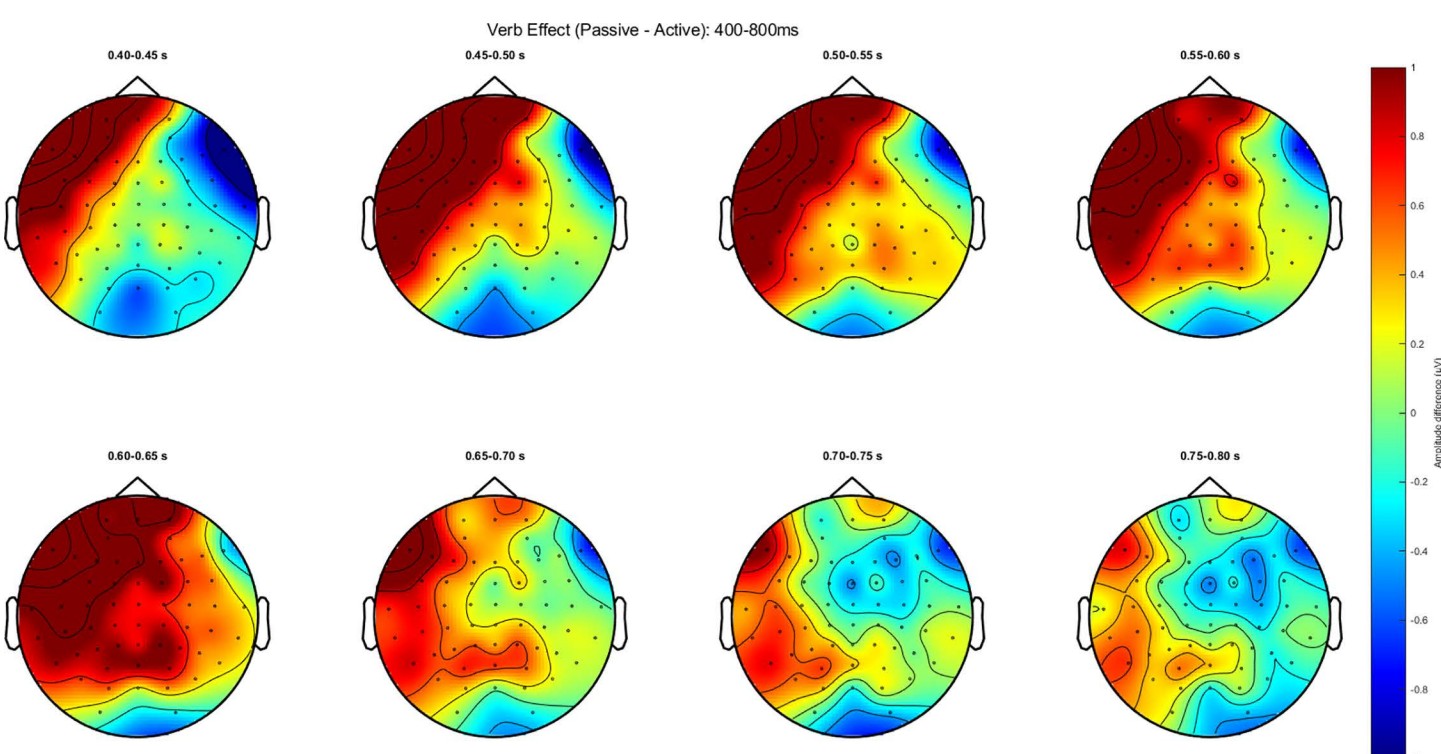

**Fig 6. Topographic plot at the NP2 for the 400 to 800ms time window (passive-active).**

complexity [62]. This may suggest that the additional cognitive load from processing passive sentences may not stem from reanalysis and repair.

One possibility is that the effect may be driven by the adoption of a heuristic strategy where the first noun is automatically assigned the role of the agent [2], and the reading of a noncanonical sentence requires revision of the automatic assignment of the agent role to the first noun phrase. This idea was also put forward by Jackson, Lorimor, & van Hell [21] whose findings are quite similar to ours. However, this explanation may not stand for Standard Indonesian: speakers of Standard Indonesian encounter passive sentences relatively more frequently, and therefore, in accordance with theories that incorporate heuristic strategies and usage-based models, the agent-first bias should not be as prominent as in other languages. Moreover, the verbs in this study are specifically chosen to be either balanced in frequency between its usage in active and passive sentence contexts or slightly used more in passives. An effective heuristic strategy based on usage frequency for speakers of SI should not involve an overwhelming agent-first bias.

On the other hand, our effect on the verb does seem to be smaller compared to the effects found in Jackson et al. [21], both visually and from our statistical tests: though in the exploratory analysis on the specific P600 time window we did find significant clusters, the pre-registered whole-epoch analysis did not observe significant effects on the verb. Assuming this is an ERP parallel to that of [21], which we think is an accurate description of the findings based on the distinct spatial distribution of the P600, attenuation of the P600 amplitude may be the result of the difference in syntactic frequency of agent-first structures in SI and English, with the latter being extremely frequent compared to its non-canonical counterparts.

Our findings contribute to the debate about universal versus language-specific factors in sentence processing. Despite Indonesian's frequent passive structures, we observed processing costs associated with passives, supporting the S/A preference hypothesis[15]. However, the somewhat smaller P600 effect (in that we did not find the effect to be significant at the verb for the whole epoch analysis) compared to English [21] suggests language-specific frequency factors may modulate this universal bias. The positivity at NP2 may reflect processing costs specific to Indonesian: given the prevalence of agentless passives, readers may initially interpret a passive-marked verb as introducing an agentless construction, requiring revision when encountering an explicit agent.

## Conclusion

The assignment of thematic roles is a vital process included in various models of language processing [2,20]. The present study teases apart the confounding effect of frequency, which is present in all previous studies investigating various word orders, and the processing of thematic roles. We examined the nature of how thematic roles are processed during real-time language processing, and whether passive sentences require additional cognitive effort to parse. While our results are not fully conclusive, there seems to be a disconnect between the ERP findings of the present study and the plethora of behavioural evidence [31–33,61] showing how passives are acquired early, used frequently, and preserved in aphasic individuals in SI.

This may be explained by the fact that the studies which looked at passives in SI (or in other Austronesian languages that have frequent passives), generally used accuracy as a measure. With unlimited time to read, this may not truly reflect the difficulty or effort required to effectively parse passives for SI speakers. Conversely, our findings also seem to be affected by the usage frequency of the passives, as the ERP effect on the verb seem to be somewhat attenuated. While this study controlled for verb-level frequency, a limitation is that it did not assess predictability within the sentence context. Future work measuring surprisal or cloze probability of verbs following specific noun phrases could directly test whether contextual predictability modulates the observed ERP effects. This approach may help explain the somewhat attenuated P600 at the verb and the sustained positivity at NP2, which can offer a more granular view on the relationship between frequency-based expectations (e.g. high frequency of agentless passives) and the agent-first preference during sentence comprehension.

## Supporting information

**S1 File.  S1 Table.** Verb frequency list.
(DOCX)

**S2 File.  S1 Fig. Raster plot showing which data points were included in the permutation test for the whole epoch at NP2.** S2 Fig. ERP (electrode P7) at the verb. S3 Fig. ERP (electrode P7) at NP2. S4 Fig. Topographic plot at the NP2 for the 500–700ms time window (passive-active). S5 Fig. Topographic plot at the Verb for the whole epoch (passive-active). S6 Fig. Topographic plot at the NP2 for the whole epoch (passive-active). S7 Fig. ERP plots at the verb, part 1 (passive-active). S8 Fig. ERP plots at the verb, part 2 (passive-active). S9 Fig. ERP plots at the verb, part 3 (passive-active). S10 Fig. ERP plots at the verb, part 4 (passive-active). S11 Fig. ERP plots at the NP2, part 1 (passive-active). S12 Fig. ERP plots at the NP2, part 2 (passive-active). S13 Fig. ERP plots at the NP2, part 3 (passive-active). S14 Fig. ERP plots at the NP2, part 4 (passive-active).
(DOCX)

## Author contributions

**Conceptualization:** Bernard A. J. Jap.

**Data curation:** Bernard A. J. Jap, Stephen Politzer-Ahles.

**Formal analysis:** Bernard A. J. Jap, Stephen Politzer-Ahles.

**Funding acquisition:** Bernard A. J. Jap, Yu-Yin Hsu, Stephen Politzer-Ahles.

**Investigation:** Bernard A. J. Jap, Yu-Yin Hsu, Stephen Politzer-Ahles.

**Methodology:** Bernard A. J. Jap, Yu-Yin Hsu, Stephen Politzer-Ahles.

**Project administration:** Bernard A. J. Jap, Yu-Yin Hsu.

**Resources:** Yu-Yin Hsu, Stephen Politzer-Ahles.

**Software:** Yu-Yin Hsu, Stephen Politzer-Ahles.

**Supervision:** Bernard A. J. Jap, Yu-Yin Hsu, Stephen Politzer-Ahles.

**Validation:** Bernard A. J. Jap, Yu-Yin Hsu, Stephen Politzer-Ahles.

**Visualization:** Bernard A. J. Jap, Yu-Yin Hsu, Stephen Politzer-Ahles.

**Writing – original draft:** Bernard A. J. Jap, Yu-Yin Hsu, Stephen Politzer-Ahles.

**Writing – review & editing:** Bernard A. J. Jap, Yu-Yin Hsu, Stephen Politzer-Ahles.

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
