## [Decision Letter · Decision Letter 0]

17 Sep 2024

PONE-D-24-16699Registered Report: Neural correlates of thematic role assignment for passives in Standard IndonesianPLOS ONE

Dear Dr. Jap,

Thank you for your continued patience throughout the review process. After a careful review of the manuscript and the detailed feedback provided by the reviewers, I have reached a decision regarding your submission. As you will notice, the reviewers’ opinions are somewhat divided. While Reviewer 2 is largely positive and suggests only minor revisions, Reviewer 1 has raised significant concerns, particularly regarding the study’s experimental design and the interpretation of the ERP data. Specifically, Reviewer 1 points out that the ERP obtained could reflect the processing of both the target word and the preceding word, which is considered as a critical limitation. Additionally, Reviewer 1 has suggested several areas where the clarity and detail within the introduction and discussion sections could be improved.

After considering both reviewers’ comments and my own assessment of the manuscript, I have decided to request a major revision. This decision is based on the following considerations:

 Given that the study was pre-registered, I believe it would be unreasonable to request significant changes to the experimental design at this stage. As far as I am aware, a stimulus onset asynchrony (SOA) of 600 ms is not uncommon in ERP studies of sentence processing. And in this study, the sentences in the active and passive conditions were matched, which should somewhat mitigate the problems of carry-over responses form the previous words.I appreciate that this study focuses on Indonesian, an under-represented language in psycholinguistic research. This adds value to the manuscript, and I recognize that no study is without its limitations.

That said, it is crucial to address Reviewer 1’s concerns carefully. In your revision, please respond comprehensively to the points raised. If you believe certain issues cannot be fully resolved, please discuss them openly as limitations of the study.

Additionally, I would like to see further elaboration on your decision to adopt a p-level of 0.1 for the statistical analysis. While the manuscript currently states that “the rationale for using a cluster threshold of p < 0.1 was to make the test more sensitive to weak, sustained effects,” there is growing concern within the scientific community about the risk of false positives and the replicability of findings. In your revision, I encourage you to provide a more detailed justification for how you balanced the aim of increasing statistical power with the need to minimize Type I errors.

I look forward to your revised submission. After receiving your revision, I will make the final decision as quickly as I can. Please do not hesitate to contact me if you have any questions.

We look forward to receiving your revised manuscript.

Kind regards,

Yiu-Kei Tsang

Academic Editor

PLOS ONE

Additional Editor Comments (if provided):

Reviewers' comments:

Reviewer's Responses to Questions

**Comments to the Author**

1. Does the manuscript adhere to the experimental procedures and analyses described in the Registered Report Protocol?

If the manuscript reports any deviations from the planned experimental procedures and analyses, those must be reasonable and adequately justified.

Reviewer #1: Yes

Reviewer #2: Yes

2. If the manuscript reports exploratory analyses or experimental procedures not outlined in the original Registered Report Protocol, are these reasonable, justified and methodologically sound?

A Registered Report may include valid exploratory analyses not previously outlined in the Registered Report Protocol, as long as they are described as such.

Reviewer #1: Yes

Reviewer #2: Yes

3. Are the conclusions supported by the data and do they address the research question presented in the Registered Report Protocol?

The manuscript must describe a technically sound piece of scientific research with data that supports the conclusions. The conclusions must be drawn appropriately based on the research question(s) outlined in the Registered Report Protocol and on the data presented.

Reviewer #1: Partly

Reviewer #2: Yes

4. Have the authors made all data underlying the findings in their manuscript fully available?

Reviewer #1: No

Reviewer #2: Yes

5. Is the manuscript presented in an intelligible fashion and written in standard English?

Reviewer #1: Yes

Reviewer #2: Yes

6. Review Comments to the Author

Please use the space provided to explain your answers to the questions above. (Please upload your review as an attachment if it exceeds 20,000 characters)

Reviewer #1: This manuscript presents an EEG study on the parsing of thematic roles during sentence comprehension in Standard Indonesian. It is very exciting to read such a study that expands our view beyond the languages that we read about all the time, having the potential to contribute to the diversification of psycho- and neurolinguistics. Unfortunately, there are a number of points that will need to be addressed to be able to publish this manuscript (if they can be addressed). The data and scripts were not accessible for review.

Main points:

* I will start with the most severe point, which I consider a fundamental flaw in the procedure: The visual presentation time of the words was 500 ms, with 100 ms blank screen between words. This means that each word can be processed for 600 ms (500 + 100) before the next word occurs. This already includes the 100 ms blank screen which goes in hand with visually evoked ERPs from the luminosity change. This means that (i) the epochs of -200 to 1000 ms relative to word onset include 100 ms of the previous word (-100 to -200 ms) as well as 400 ms of the next word (600 to 1000 ms) and (ii) the 500-700 ms window where a positivity is being found is entirely where the word is not being displayed anymore and also includes 100 ms of the next word. In addition, 500 ms may be a quite short presentation time, given that Indonesian words (esp. the verbs) are probably longer than average words in English. I am thus not confident whether the results are even interpretable, especially in terms of a P600-like effect.

* The presentation of results is extremely lacking. ERPs from only one electrode are shown and topographic maps are only shown for the 500-700 ms of NP2. ERP plots for all electrodes as well as topographic maps for the whole epochs (e.g. in 100 ms or 50 ms steps) need to be shown for all critical words. Otherwise, the readers have no way to assess the results in any meaningful way.

* The manuscript lacks in embedding in currrent theoretical discussions on sentence comprehension. There is no discussion of incremental processing theories from which predictions could be derived. The predictions of the experiment and the reasons why passives (in Indonesian) should lead to reanalysis responses are not clear. Actually, given the high frequency of agentless passives (which is mentioned for the first time on the second-to-last page!) could predict that disambiguation towards agents would require analysis. In addition, any discussion is missing on where the expectations for potential effects/mechanisms of thematic role assignment in incremental comprehension are coming from, particularly the agent preference (cf. works by Bornkessel-Schlesewsky and also 10.1371/journal.pone.0132819 and 10.1016/j.jml.2008.02.003). This becomes particularly apparent on p. 23 where the predictions are being presented. For the first tome a transitivity expectation is being mentioned and also that the agent is the default role is not being motivated. An integration in the current discussion on the need for more cross-linguistic studies is also missing. Overall, the impression arises that the current literature was not well integrated into the manuscript.

* There have been recent studies that exactly addressed the question that the manuscript claims to be untackled, namely the role of frequency in thematic role processing (DOIs: 10.1162/nol_a_00121, 10.1111/cogs.13340, 10.1080/23273798.2023.2250023). These studies used corpus- and Large-Language-Model-derived probabilities and surprisal measures to explore how frequency and general processing heuristics interact in several languages. The current manuscript falls severely short in it's attempts to do the same, compared with these publications. These studies (and further ones) also consistently found N400 effects for disambiguation towards patient readings, which is being interpreted as the detection of a prediction error. These previous studies, however, do not invalidate the current attempt, but they should be discussed and put in relation to the experiment reported here.

* The operationalisation of frequency unfortunately does not cover the relevant aspects for the sentence comprehension processes under investigation. First, a lot of emphasis is being put on passives being more frequent than in English. But if actives are still more frequent overall, then this does not resolve the frequency problem. In principle, based on statistical learning and experiences with their language, comprehenders would also be sensitive to a very small imbalance of structural frequencies. Second, the way of controlling for frequency by determining the verbs' biases to occur as actuves or passives does not capture the relevant aspects of frequency fo incremental role processing. When the first NP is encountered, the parser will assign a role to it (generally assumed to be the agent, based on frequency *and* the agent preference). At this point in time, when the role expectation is being formed, nothing about the verb is known. Thus, what would capture the frequency aspect better would be the surprisal or the cloze probability of an active or passive verb following th specific nouns or of the specific verbs with these specific forms following the nouns.

* What exactly is the disambiguating region in the current sentences? I first thought it would be the verb, but in the discussion when agentless passives are being mentioned, it sounds like only the combination of verb and NP2 fully disambiguates the role of NP1. If that is the case, then this could partially explain why there is no clear ERP pattern the verb. At the same time, the information in the verb (both the lexical verb and the active/passive marking) must contribute to forming an interpretation of the sentence (as it is also being described on p. 11).

Other points:

# Abstract #

The abstract mentioines that previous studies did not yet identify a unique correlate of thematic role processing. Is there any reason that there has to be an ERP that is *unique* to thematic role processing? In the context of seeing language processing as a cognitive skill, possibly relying on domain-general processes, this view does not become clear to me.

What does "syntactically more complicated" mean? Is there a more accurate way to describe passives?

The abstract also says that "the non-canonical word order is common" in Standard Indonesian. If it is common (and as it turns out in the discussion, even more frequent), why would the passive then be non-canonical? Just because the word order in English is canonically SVO, this does not have any automatic consequences for other languages. Each language should rather be treated on its own terms. Currently, Standard Indonesian is being treated as if it is just kind of like English, while it's system of actives and passives is quite different.

# Introduction #

Throughout the manuscript, it does not become very clear what level of linguistic analysis this is about. The terminology switches between thematic roles, subject and object and canonical vs. non-canonical word orders. I propose to reframe everything in terms of parsing (predicting) thematic/semantic roles, making the other distinctions more subordinate. This also applies to Table 1; is this paper about subject-object or about agent-patient (they ofter overlap, but not always)? I recommend replacing "SVO" with "agent-verb-patient" throughout the manuscript, also to avoid pressing Indonesian into the procrustean bed of English grammar terms.

The claim that there is very little research on sentences without anomalies (p. 4) seems to not be right. There are tons of studies with non-violated sentences.

On p.4 at the discussion of example 1, one could also argue that "the man" is the agent of the main clause verb (admitted) and it is rather the realisation that there is a relative clause that leads to processing difficulties. Could that be another possible explanation for this example?

The case marking in the glosses for example 2 needs to be on the determiner or on the whole NP, for that matter, but not on the noun.

Why is the sentence in 5a described as canonical (p. 10) when it is actually a less frequent structure than 5b?

The abbreviation SI needs to be introduced in the main text before being used.

A discussion of the functional role of P600 and N400 in sentence comprehension would be useful to situate the current experiment in the literature and to interpret the findings. Why is it expected that disambiguation towards a passive should elicit a P600? Relevant papers are (among others): 10.1080/01690960902965951, 10.1111/cogs.12461, 10.3389/fnhum.2013.00758, 10.1111/psyp.13351, 10.1371/journal.pone.0257430, 10.1097/00001756-200110290-00048, 10.3389/fpsyg.2019.00298

# Method #

It is hard to believe that the largest study in the literature on sentence comprehension has 25 participants. Haupt et al. (2008), for example had 40 participants (10.1016/j.jml.2008.02.003).

In the discussion about what baseline to use, the paper by Alday on "How much baseline correction do we need in ERP research?" (10.1111/psyp.13451) is missing. Not only because it is more recent but also because it provides a convincing solution for the problem.

# Materials #

The use of inanimate NPs would have been beneficial, too, because in combination with the verb, these could have been easier to disambiguate.

What are the units in Table 4?

Figure 1 should connect the dots of each verb in the Active and Passive columns so that the reader can see which points on each side belong together. In ggpplot, this can be done with geom_path().

# Procedure #

The comprehension questions should have been randomly distributed over all trials. By just presenting them every 3-8 trials, participants knew that after they answered a comprehension question, they would not have to answer another one for at least 2 more trials. How could that have affected their comprehension strategies? What if they zoned out for the next 2 sentences?

Asking comprehension questions only for a part of the critical trials also means that only for a part of those there is information about comprehension success.

The comprehension data should be presented, at least descriptively.

p. 19: Was the average of the mastoids used as the reference electrode?

Restricting the removal of independent components to only 4, when there are 64 electrodes, appears to be unnecessarily conservative.

# EEG recording and preprocessing #

The Bad Channels (2) criterion says that every participant with a cluster of bad electrodes would be excluded. Just as food for thought, would this also have been the case if in the last trial of the experiment, such a cluster emerges that all the previous data would be excluded? Some kind of time criterion for this would have been good. (I know that it didn't apply here, but for future studies.)

# Statistical analysis #

I know that this is a registered report, so the analyses were approved beforehand. However, cluster-based permutation approaches by virtue of being univariate tests don't allow to include (a) by-item variation, which is usually also high in language comprehension tasks (just look at the spread of your acceptability ratings) and (b) frequency as a predictor. Regression-based approaches would allow to do that.

Even though the frequency classes were matched for active and passive, there is still considerable spread as can be seen in Figures 1 and 2. Sassenhagen and Alday show that controlling for nuisance variables by running a stastical significance test beforehand actually does not achieve this goal (10.1016/j.bandl.2016.08.001). As an attempt to *partially* cover this problem, the data for each participant could be split into 2 bins (based on Figure 2) using the uvar attribute in FieldTrip's permutation tests (https://www.fieldtriptoolbox.org/faq/how_can_i_use_the_ivar_uvar_wvar_and_cvar_options_to_precisely_control_the_permutations/).

# Predictions #

Prediction 3 (a lack of ERP correlates) cannot be tested in the current framework because frequentist statistics cannot distinguish between genuine null effects and a lack of statistical power.

p. 24: While the study by Jackson et al. is very similar to the current study, other studies such as 10.1016/j.jml.2008.02.003 and 10.1371/journal.pone.0132819 are also very similar.

What does it mean that "verbs in Basque and Japanese provide no additional thematic information" (p. 24) given that the verbs provide the event the sentence is being about and also determine the argument structure, so basically disambiguate the roles if it has not been done?

# Results #

On p. 26, it is mentioned that a difference is not significant on the verb (p = .087) but on p. 22 it is mentioned that the threshold for signficance is set to p < .1.

Reviewer #2: The authors may add a conclusion section.

"...P600 time window which are maximally..." in line four, page 15 should probably be " ... P600 time window which is maximally...

"... the participants are similar to ..." in line seven, page 15 should probably be "... the participants is similar to ..."

7. PLOS authors have the option to publish the peer review history of their article (what does this mean? ). If published, this will include your full peer review and any attached files.

**Do you want your identity to be public for this peer review?** For information about this choice, including consent withdrawal, please see our Privacy Policy .

Reviewer #1: No

Reviewer #2: No

---

## [Author Response · Author response to Decision Letter 1]

17 Mar 2025

We have attached the response to reviewer document (please view the responses in the document). Below is the pasted version, but with no formatting.

Thank you for your feedback and the opportunity to revise our manuscript. In the following response letter and in our revised manuscript, we will try to address as many of the comments as are possible within the scope of this revision.

As this is a Stage 2 registered report, and per PLoS ONE’s policy (https://journals.plos.org/plosone/s/what-we-publish#loc-registered-reports) the study at this stage should adhere to the approved Stage 1 protocol; if other aspects of the study are reviewed at this stage, then PLoS ONE does not actually have registered reports in any meaningful sense.

Nevertheless, we share the editor’s and reviewers’ interest in making the most accurate conclusion possible on the basis of the available data, and thus we have revised the manuscript to more transparently address potential limitations of the approved protocol and improve clarity. The changes in the main manuscript file are marked with track changes.

Editor:

Additionally, I would like to see further elaboration on your decision to adopt a p-level of 0.1 for the statistical analysis. While the manuscript currently states that “the rationale for using a cluster threshold of p < 0.1 was to make the test more sensitive to weak, sustained effects,” there is growing concern within the scientific community about the risk of false positives and the replicability of findings. In your revision, I encourage you to provide a more detailed justification for how you balanced the aim of increasing statistical power with the need to minimize Type I errors.

We have clarified in the manuscript that the p<.1 criterion for cluster identification is not the same as the alpha level for the actual statistical test. Generating clusters is an intermediate step in cluster-based permutation analysis, and is not when statistical inference actually occurs. The actual statistical test (where we make conclusions about whether a difference between conditions is significant) uses a standard p<.05 threshold. The clarification is on page 25-26: “Effects were treated as significant if they obtained a p-value below .05 for the permutation test. The abovementioned p<.1 criterion was only used in the procedure for generating clusters, not for actually evaluating the overall significance of the test. The p threshold used for generating clusters does not affect the conservativity of the overall test, as explained by [59]”

Reviewer #1: This manuscript presents an EEG study on the parsing of thematic roles during sentence comprehension in Standard Indonesian. It is very exciting to read such a study that expands our view beyond the languages that we read about all the time, having the potential to contribute to the diversification of psycho- and neurolinguistics. Unfortunately, there are a number of points that will need to be addressed to be able to publish this manuscript (if they can be addressed). The data and scripts were not accessible for review.

Main points:

* I will start with the most severe point, which I consider a fundamental flaw in the procedure: The visual presentation time of the words was 500 ms, with 100 ms blank screen between words. This means that each word can be processed for 600 ms (500 + 100) before the next word occurs. This already includes the 100 ms blank screen which goes in hand with visually evoked ERPs from the luminosity change. This means that (i) the epochs of -200 to 1000 ms relative to word onset include 100 ms of the previous word (-100 to -200 ms) as well as 400 ms of the next word (600 to 1000 ms) and (ii) the 500-700 ms window where a positivity is being found is entirely where the word is not being displayed anymore and also includes 100 ms of the next word. In addition, 500 ms may be a quite short presentation time, given that Indonesian words (esp. the verbs) are probably longer than average words in English. I am thus not confident whether the results are even interpretable, especially in terms of a P600-like effect.

We do not see any way the visual presentation could cause spurious differences between conditions, since there is no systematic difference between the conditions on the next word, especially one that would occur very early on (sub 100ms – which would be something low-level such as visual/orthographic differences), therefore, any difference that is captured during the 500-700ms time window is almost certainly a result of the overall differences between the conditions (e.g. verb morphology/word order) or the original word stimuli instead of the response to the following word at the 0-100ms.

Furthermore, the stimulus timing used in the present study is typical of sentence processing experiments using rapid-serial-visual presentation, and if anything was longer than usual. We used a longer SOA because of the relatively long words in Standard Indonesian. For comparison, Jackson et al (2020) for English uses 350ms per word with 100ms ITI (450 ms SOA); Matzke et al (2000) for German uses 200ms per word with 300ms ITI (500ms SOA) between word onsets; Wolff et al (2008) for Japanese, presents multiple words (bunsetsu) per screen using 650ms with 100ms ITI (750 ms SOA, but for multiple words); and Erdocia et al (2009) for Basque uses 250ms per word with 250ms ITI (500 ms SOA).

To further address this, we have added clarification in the Procedure that our presentation timing is standard in sentence processing studies using RSVP and that it does not compromise the interpretability of our results.

* The presentation of results is extremely lacking. ERPs from only one electrode are shown and topographic maps are only shown for the 500-700 ms of NP2. ERP plots for all electrodes as well as topographic maps for the whole epochs (e.g. in 100 ms or 50 ms steps) need to be shown for all critical words. Otherwise, the readers have no way to assess the results in any meaningful way.

We appreciate the reviewer's feedback on our ERP results presentation. We've improved the presentation by adding 3x3 ERP plots for both conditions and extended topographic maps (100ms steps) to the main paper. As requested, plots of all recorded electrodes (64 channels) in all conditions as well as the topographic plots of the whole epoch (including baseline) are now included in the S4 appendix (Supporting Information).

* The manuscript lacks in embedding in currrent theoretical discussions on sentence comprehension. There is no discussion of incremental processing theories from which predictions could be derived. The predictions of the experiment and the reasons why passives (in Indonesian) should lead to reanalysis responses are not clear. Actually, given the high frequency of agentless passives (which is mentioned for the first time on the second-to-last page!) could predict that disambiguation towards agents would require analysis. In addition, any discussion is missing on where the expectations for potential effects/mechanisms of thematic role assignment in incremental comprehension are coming from, particularly the agent preference (cf. works by Bornkessel-Schlesewsky and also 10.1371/journal.pone.0132819 and 10.1016/j.jml.2008.02.003). This becomes particularly apparent on p. 23 where the predictions are being presented. For the first tome a transitivity expectation is being mentioned and also that the agent is the default role is not being motivated. An integration in the current discussion on the need for more cross-linguistic studies is also missing. Overall, the impression arises that the current literature was not well integrated into the manuscript.

Thank you for pointing this out. We have integrated current theories of incremental processing and thematic role assignment into our manuscript, with particular attention to the S/A preference hypothesis (Bickel et al., 2015) and argument linking processes (Haupt et al., 2008). We've added discussion of incremental processing theories (p.5), clarified that Indonesian has frequent agentless passives (p.12), revised our predictions based on these frameworks (p.26-27), and integrated these concepts into our discussion of findings (p.31-32).

* There have been recent studies that exactly addressed the question that the manuscript claims to be untackled, namely the role of frequency in thematic role processing (DOIs: 10.1162/nol_a_00121, 10.1111/cogs.13340, 10.1080/23273798.2023.2250023). These studies used corpus- and Large-Language-Model-derived probabilities and surprisal measures to explore how frequency and general processing heuristics interact in several languages. The current manuscript falls severely short in it's attempts to do the same, compared with these publications. These studies (and further ones) also consistently found N400 effects for disambiguation towards patient readings, which is being interpreted as the detection of a prediction error. These previous studies, however, do not invalidate the current attempt, but they should be discussed and put in relation to the experiment reported here.

Thank you for highlighting these important recent papers. We have incorporated discussion of Huber et al. (2024), Sauppe et al. (2023), and Isasi-Isasmendi et al. (2024) into our manuscript in 'The present study' section.

* The operationalisation of frequency unfortunately does not cover the relevant aspects for the sentence comprehension processes under investigation. First, a lot of emphasis is being put on passives being more frequent than in English. But if actives are still more frequent overall, then this does not resolve the frequency problem. In principle, based on statistical learning and experiences with their language, comprehenders would also be sensitive to a very small imbalance of structural frequencies. Second, the way of controlling for frequency by determining the verbs' biases to occur as actuves or passives does not capture the relevant aspects of frequency fo incremental role processing. When the first NP is encountered, the parser will assign a role to it (generally assumed to be the agent, based on frequency *and* the agent preference). At this point in time, when the role expectation is being formed, nothing about the verb is known. Thus, what would capture the frequency aspect better would be the surprisal or the cloze probability of an active or passive verb following th specific nouns or of the specific verbs with these specific forms following the nouns.

Regarding your first point about the relative frequency of active vs. passive structures within Indonesian: We have revised our manuscript to more clearly emphasize the relative frequencies of active and passive constructions within Indonesian itself. While active constructions do remain somewhat more frequent overall in Indonesian (as in virtually all languages), the critical aspect for our study is that the frequency differential between active and passive in Indonesian is much smaller than in previously studied languages. As noted on the paper, passive verbs account for 30-40% of all verb tokens in Indonesian text (compared to just 9% in English), with passive input frequency of 28-35% (compared to 4-5% in English).

Regarding your second point about incremental processing at NP1 before the verb is encountered: We agree that examining surprisal or cloze probability of active/passive verbs following specific nouns would provide insights into incremental role processing. However, as you rightly note, our study focuses specifically on the disambiguation point (the verb) and subsequent processing. Since our critical regions are the verb itself and NP2, the verb information is already available at the points we're measuring. We've added a remark in the Discussion section acknowledging this and suggesting it as a direction for future research.

* What exactly is the disambiguating region in the current sentences? I first thought it would be the verb, but in the discussion when agentless passives are being mentioned, it sounds like only the combination of verb and NP2 fully disambiguates the role of NP1. If that is the case, then this could partially explain why there is no clear ERP pattern the verb. At the same time, the information in the verb (both the lexical verb and the active/passive marking) must contribute to forming an interpretation of the sentence (as it is also being described on p. 11).

The primary disambiguating region in our sentences is the verb, where the voice marking (men- prefix for active, di- prefix for passive) immediately indicates the thematic role of NP1. However, as we discuss later in the paper, the prevalence of agentless passives in Indonesian creates an interesting secondary effect at NP2. When readers encounter a passive-marked verb, they likely expect an agentless passive structure (the most frequent type), requiring additional integration processes when an explicit agent appears at NP2. Indeed, this expectation pattern could contribute to the stronger ERP effect we observed at NP2 compared to the verb.

Other points:

# Abstract #

The abstract mentioines that previous studies did not yet identify a unique correlate of thematic role processing. Is there any reason that there has to be an ERP that is *unique* to thematic role processing? In the context of seeing language processing as a cognitive skill, possibly relying on domain-general processes, this view does not become clear to me.

What does "syntactically more complicated" mean? Is there a more accurate way to describe passives?

The abstract also says that "the non-canonical word order is common" in Standard Indonesian. If it is common (and as it turns out in the discussion, even more frequent), why would the passive then be non-canonical? Just because the word order in English is canonically SVO, this does not have any automatic consequences for other languages. Each language should rather be treated on its own terms. Currently, Standard Indonesian is being treated as if it is just kind of like English, while it's system of actives and passives is quite different.

We're describing the current literature state where inconsistent ERP components have been found across studies. We completely agree – there does not have to be one unique component that correlates to thematic role processing, and we are very much open to the possibility that there is not one. But one cannot conclude that there is no such ERP correlate unless research has robustly tested for it and failed to find it. Our research contributes to this by investigating whether the inconsistencies in previous attempts to find such an ERP correlate stem from confounding frequency effects.

We’ve also revised the abstract: "syntactically more complicated” to "sentences with a ‘non-default’ word order" for clarity. "non-canonical word order is common" has been revised to "a language where passive structures occur with high frequency and are comparable in frequency to active structures"

# Introduction #

Throughout the manuscript, it does not become very clear what level of linguistic analysis this is about. The terminology switches between thematic roles, subject and object and canonical vs. non-canonical word orders. I propose to reframe everything in terms of parsing (predicting) thematic/semantic roles, making the other distinctions more subordinate. This also applies to Table 1; is this paper about subject-object or about agent-patient (they ofter overlap, but not always)? I recommend replacing "SVO" with "agent-verb-patient" throughout the manuscript, also to avoid pressing Indonesian into the procrustean bed of English grammar terms.

We’ve replaced the SVO terms with agent-verb-patient throughout the manuscript when referring to the current study/SI. We retain some use of subject or object when referring to certain sentence types (e.g. object relatives), as this was how the studies we cited have referred to it. We’ve also edited Table 1.

The claim that there is very little research on sentences without anomalies (p. 4) seems to not be right. There are tons of studies with non-violated sentences.

We’ve edited this part (beginning of the subsection Word order processing in ERP re

---

## [Editor Report · Decision Letter 1]

21 Mar 2025

Registered Report: Neural correlates of thematic role assignment for passives in Standard Indonesian

PONE-D-24-16699R1

Dear Dr. Jap,

Thank you for submitting the revised version of your manuscript. After carefully reading the manuscript myself, I am pleased to inform you that your manuscript has been accepted for publication without the need for an additional round of review. This decision is based on the following reasons:

1. Although Reviewer 1 previously expressed concerns regarding the methodology, I recognize the importance of adhering to the pre-registered protocol. You have made every effort to follow the approved protocol, and any deviations have been clearly documented.

2. Your revised manuscript provides adequate justification for the chosen methodological details.

3. Reviewer 2 expressed satisfaction with your manuscript.

Your manuscript has been judged scientifically suitable for publication and will be formally accepted for publication once it meets all outstanding technical requirements.

Kind regards,

Yiu-Kei Tsang

Academic Editor

PLOS ONE
---

## [Editor Report · Acceptance letter]

PONE-D-24-16699R1

PLOS ONE

Dear Dr. Jap,

I'm pleased to inform you that your manuscript has been deemed suitable for publication in PLOS ONE. Congratulations! Your manuscript is now being handed over to our production team.

Kind regards,

on behalf of

Dr. Yiu-Kei Tsang

Academic Editor

PLOS ONE